# Prevalence and risk factors of disordered eating risk among Yemeni medical students: A national multicenter cross-sectional study

Naji Al-bawah[1,☯,*], Mohamed Baklola[2,☯,*], Anas H. Khalifeh[3], Ahmed Al-Eryani[1],
Anas Zakarya Nourelden[4], Saif Alaribi[5], Ehab Sharyan[1], Fayda Al-wesabi[6],
Aaida Al Wesabi[6], Ahdab Al-Jaberi[1], Sama Shamsan[1], Ahmed Abdulmughni[1],
Ghailan Al-Jarbani[1], Hisham Salman[1], Amira Yasmine Benmelouka[7]

**1** Faculty of Medicine, Sana'a University, Sana'a, Yemen, **2** Faculty of Medicine, Mansoura University, Mansoura, Egypt, **3** Department of Community and Mental Health Nursing, Faculty of Nursing, Zarqa University, Zarqa, Jordan, **4** Faculty of Medicine, Al-Azhar University, Cairo, Egypt, **5** Faculty of Medicine, Emirates International University, Sana'a, Yemen, **6** Faculty of Medicine, 21 September University, Sana'a, Yemen, **7** Faculty of Medicine, University of Algiers, Algiers, Algeria

☯ These authors contributed equally to this work.
\* mohamedbaklola2000@gmail.com (MB); najialbawah@gmail.com (NA)

## Abstract

### Background

Disordered eating behaviors represent an important mental health concern among medical students, who are exposed to substantial academic and psychological stressors. Data on disordered eating risk in Yemen remain limited. This study aimed to estimate the prevalence of disordered eating risk among Yemeni medical students and to examine associated sociodemographic, lifestyle, medical, and psychological factors.

### Methods

A national multicenter cross-sectional study was conducted between June and July 2025 across fifteen Yemeni medical colleges. Undergraduate students completed a structured, self-administered online questionnaire using a convenience sampling approach. Disordered eating risk was assessed using the Eating Attitudes Test (EAT-26), with a cutoff score of ≥20. Additional items assessed sociodemographic characteristics, lifestyle factors, chronic medical conditions, and self-reported psychiatric diagnoses. Multivariable logistic regression was performed to examine factors associated with disordered eating risk.

### Results

A total of 3,540 students participated. Overall, 30.4% screened positive for disordered eating risk. Disordered eating risk was associated with food insecurity, parental

**Data availability statement:** The datasets generated and analyzed during this study are publicly available in the Figshare repository at https://doi.org/10.6084/m9.figshare.30987517.

**Funding:** The author(s) received no specific funding for this work.

**Competing interests:** The authors have declared that no competing interests exist.

perception of being overweight, weight dissatisfaction, type 1 diabetes mellitus, past suicide attempts, and several self-reported psychiatric conditions, including post-traumatic stress disorder, obsessive compulsive disorder, social anxiety disorder, bipolar disorder, schizophrenia, and major depressive disorder. Weight satisfaction was associated with lower odds of risk. Additionally, 31.6% of participants reported a prior self-reported diagnosis of an eating disorder, which was not independently verified.

## Conclusion

Disordered eating risk was common among Yemeni medical students and was associated with multiple sociodemographic, medical, and psychological factors. Given the cross-sectional design and reliance on self-reported data, causal relationships cannot be established. The findings suggest that consideration may be given to screening and supportive mental health resources within medical training institutions.

## Background

Eating disorders (EDs) are complex psychiatric conditions marked by persistent disturbances in eating behaviors and an excessive focus on body weight and shape, resulting in significant impairment in mental, physical, and psychosocial functioning [1,2]. According to the Diagnostic and Statistical Manual of Mental Disorders, Fifth Edition (DSM-5), EDs include anorexia nervosa, bulimia nervosa, binge eating disorder, avoidant or restrictive food intake disorder, pica, rumination disorder, and other specified feeding or eating disorders [3]. Alongside clinically diagnosed EDs, disordered eating refers to a broader spectrum of maladaptive eating attitudes and behaviors that do not necessarily meet diagnostic criteria but are associated with substantial physical and psychological morbidity, including electrolyte disturbances, hormonal dysregulation, cardiovascular complications, and reduced bone density [4].

The development of EDs is multifactorial. Predisposing factors include genetic vulnerabilities and certain personality traits, while triggering and maintaining factors may involve environmental influences, social isolation, and maladaptive coping mechanisms [5]. Globally, the prevalence of EDs is estimated at 5.7% among females and 2.2% among males [6]. However, screening-based estimates of disordered eating risk appear substantially higher in specific subgroups, particularly medical students. A recent systematic review and meta-analysis reported a pooled prevalence of 10.4% among medical students based largely on screening instruments [7], while other individual studies suggest rates reaching 15% [8], 28.6% in Jordan [9], and specific ED diagnoses such as binge eating and bulimia nervosa reported at 5% and 1.2% respectively, in Nepal [10].

Medical students are considered particularly vulnerable to disordered eating behaviors due to a combination of academic, psychological, and social pressures. Contributing factors may include anxiety, depression, academic overload, burnout, limited coping skills, lack of social support, weight stigma, and body image dissatisfaction [11–13]. Studies also suggest that vulnerability may increase during the

transition to clinical training years [14]. Difficulties in emotional processing, including alexithymia and impaired social cognition, may further elevate this risk [15].

EDs among students are associated with diminished cognitive performance, mood disturbances, decreased motivation, and impaired academic achievement [16]. Early identification of disordered eating patterns is crucial to preventing the escalation to more severe clinical conditions [17]. Early identification of disordered eating risk is crucial to preventing progression to more severe clinical conditions [11].

Although interest in eating disorder research has grown in the Middle East and North Africa (MENA) region, significant data gaps remain [18]. A recent large-scale multinational study involving over 5,000 medical students across the MENA region reported that approximately 24.8% were at risk of eating disorders, with key risk factors including type 1 diabetes mellitus, schizophrenia, autism, female gender, inflammatory bowel disease (IBD), and frequent exposure to the thin-ideal body image [19]. Protective factors included regular physical activity and satisfaction with body weight. Despite these findings, Yemen was not included in this study, and to date, no published research has specifically examined disordered eating risk among Yemeni medical students. Given Yemen's unique sociocultural context, ongoing conflict, limited mental health resources, and increasing exposure to media-driven body ideals, there is a critical need to assess the prevalence and correlates of disordered eating risk in this population. Therefore, this study aims to estimate the prevalence of disordered eating risk among medical students in Yemen and to identify associated sociodemographic, lifestyle, medical, and psychological factors.

## Methods

### Study design

A national multicenter cross-sectional study was conducted to estimate the prevalence of disordered eating risk and to identify associated sociodemographic, lifestyle, medical, and psychological factors among medical students in Yemen. The study followed the Strengthening the Reporting of Observational Studies in Epidemiology (STROBE) guidelines for cross-sectional studies [20].

### Study setting and duration

The study was conducted across fifteen medical colleges located in different governorates of Yemen, including Sana'a, Aden, Taiz, Ibb, and Hadramout. These institutions were selected to ensure diversity in educational settings and broad geographical representation. Ethical approval was obtained on June 21, 2025. Participant recruitment and data collection were carried out from June 22, 2025, to July 22, 2025.

### Study population and participants

The target population consisted of undergraduate medical students enrolled in Yemeni medical colleges during the study period. Eligible participants were students from all academic years, ranging from first to sixth year. Inclusion criteria were current enrollment in a Yemeni medical college, age 18 years or older, and voluntary agreement to participate. Exclusion criteria included incomplete questionnaires and duplicate responses identified during data cleaning. Participation was entirely voluntary, and no financial or academic incentives were provided.

### Sampling and recruitment

A non-probability convenience sampling strategy was employed. The questionnaire was distributed electronically through official student communication platforms, including institutional mailing lists and student-managed social media groups across the participating medical colleges. Participation was voluntary, and students self-selected into the study by accessing the survey link.

 

Due to the nature of online distribution through multiple communication channels, the exact number of students who received or viewed the invitation could not be determined. Consequently, a precise response rate could not be calculated. Although the study included participants from fifteen medical colleges across different regions of Yemen, the sample may not be fully representative of all Yemeni medical students. This limitation should be considered when interpreting the findings.

## Sample size calculation

The required sample size was calculated using GPower software version 3.1.9.7, which is a widely used and validated tool for power and sample size estimation in biomedical and epidemiological research, particularly for regression-based analyses [21]. GPower allows flexible modeling of logistic regression parameters and provides reliable estimates of statistical power based on effect size assumptions derived from prior literature.

An a priori power analysis was conducted for a binary logistic regression model, given the dichotomous nature of the primary outcome variable, defined as disordered eating risk based on an EAT-26 score of 20 or higher. The analysis assumed a two-tailed test, an odds ratio of 1.3 representing a small to moderate effect size commonly reported in studies of disordered eating among medical students, a significance level of 0.05, and a desired statistical power of 0.95. The expected probability of disordered eating risk under the null hypothesis was set at 0.20, in accordance with prevalence estimates reported in previous studies of medical student populations [22].

Based on these parameters, the minimum required sample size was estimated at 1,188 participants. To account for a potential non-response rate or incomplete questionnaires of approximately 20 percent, the target sample size was increased to 1,426 participants. Ultimately, 3,540 students completed the survey, exceeding the minimum requirement and thereby enhancing the statistical precision and robustness of the findings.

## Study tools

Data were collected using a structured, self-administered online questionnaire developed for the purposes of this study. The questionnaire was distributed electronically through official student communication platforms of participating medical colleges and consisted of three main components. The first component assessed sociodemographic and lifestyle characteristics, including age, gender, academic year, marital status, smoking status, qat chewing, physical activity, self-reported height and weight, food insecurity, and parental perceptions related to body weight.

Height and weight were self-reported by participants. Body mass index (BMI) was calculated as weight in kilograms divided by height in meters squared (kg/m²). During data cleaning, anthropometric values were screened for plausibility, and biologically implausible values were excluded based on predefined acceptable ranges. The use of self-reported measurements may introduce reporting bias and should be considered when interpreting BMI-related findings.

The second component evaluated disordered eating risk using the Eating Attitudes Test, 26-item version (EAT-26), a widely used screening instrument designed to identify individuals at increased risk of disordered eating behaviors [23]. The EAT-26 comprises 26 items rated on a six-point Likert scale, with standardized scoring procedures yielding a total score ranging from 0 to 78. A cutoff score of 20 or higher was used to indicate elevated risk of disordered eating behaviors requiring further clinical assessment. The EAT-26 is a freely available tool and has demonstrated strong psychometric properties across diverse populations. In the present study, participants completed either the Arabic or English version of the scale, both of which have been previously validated and used in Middle Eastern and medical student populations. Internal consistency in the current sample was excellent, with a Cronbach alpha coefficient of 0.88 [24].

The third component of the questionnaire assessed health-related and psychological variables. Participants were asked to report any history of chronic medical conditions and whether they had ever been diagnosed by a healthcare professional with specific psychiatric conditions, including post-traumatic stress disorder, obsessive compulsive disorder, social anxiety disorder, bipolar disorder, schizophrenia, major depressive disorder, or borderline personality disorder. These

psychiatric diagnoses were self-reported and were not independently verified through medical records or clinical assessment. Participants were also asked about previous suicide attempts, body weight satisfaction, and selected personality traits relevant to disordered eating risk, including perfectionism, impulsiveness, and obsessive traits. Prior diagnosis of an eating disorder was assessed using a single self-reported item asking whether the participant had ever received a diagnosis of an eating disorder from a healthcare professional. This information was not independently verified.

### Pilot testing

Prior to the main data collection, a pilot study was conducted among 40 medical students from two Yemeni medical colleges who were not included in the final analysis. The pilot aimed to assess clarity, comprehension, cultural appropriateness, and technical functionality of the online questionnaire. Minor linguistic and formatting modifications were made based on participant feedback. The pilot data were also used to assess internal consistency, yielding a Cronbach alpha of 0.86 for the EAT-26, indicating high reliability.

### Ethical considerations

Ethical approval was obtained from the Institutional Review Board of Amran University (Approval No. 854). Electronic informed consent was obtained from all participants prior to accessing the questionnaire. Participants were informed about the study objectives, voluntary nature of participation, confidentiality of responses, and their right to withdraw at any time without consequences. All data were collected anonymously, stored securely, and accessed only by the research team.

### Statistical analysis

Data were analyzed using IBM SPSS Statistics version 28.0. Descriptive statistics were used to summarize participant characteristics. Categorical variables were reported as frequencies and percentages, while continuous variables were presented as means and standard deviations. The Kolmogorov Smirnov test was used to assess normality of continuous variables to guide descriptive presentation. Disordered eating risk was defined as an EAT-26 score of 20 or higher and was treated as a binary outcome variable. Associations between participant characteristics and disordered eating risk were initially examined using chi-square tests or Fisher's exact tests, as appropriate. Variables with a p-value less than 0.20 in bivariate analyses were entered into a multivariable logistic regression model using the enter method. A p-value threshold of 0.20 was selected to avoid excluding potentially important confounders at the preliminary stage, consistent with recommended epidemiologic modeling practices. In addition, key sociodemographic variables (including age and gender) were retained in the multivariable model regardless of statistical significance due to their theoretical relevance. Multicollinearity was assessed using variance inflation factors. Model fit was evaluated using the Hosmer–Lemeshow goodness-of-fit test. Adjusted odds ratios (ORs) with 95% confidence intervals (CIs) were reported. A two-tailed p-value less than 0.05 was considered statistically significant.

## Results

### Participant characteristics

A total of 3,540 undergraduate medical students from fifteen Yemeni medical colleges participated in the study. Based on the EAT-26 cutoff score, 1,076 participants (30.4%) were classified as having disordered eating risk, while 2,464 participants (69.6%) did not meet the cutoff. Fig 1 illustrates the self-reported history of eating disorder diagnosis among the study participants. Overall, 1,117 students (31.6%) reported having been previously diagnosed with an eating disorder, while the majority of participants (68.4%, n = 2,423) reported no prior diagnosis. The overall sample included 1,896 females (53.6%) and 1,644 males (46.4%). Participants were distributed across all academic years, with relatively comparable representation from first through sixth year.

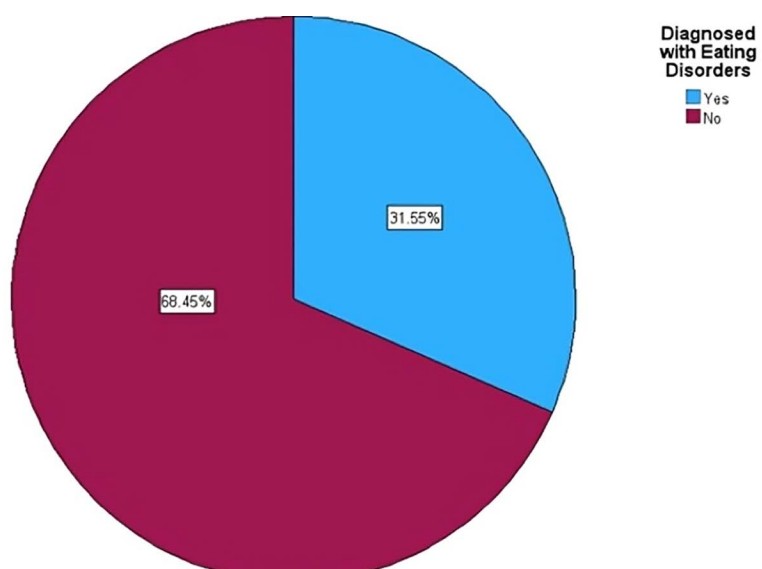

**Fig 1. Self-reported history of eating disorder diagnosis among medical students.** Proportion of participants reporting a prior diagnosis of an eating disorder. Data are based on self-report and were not independently verified. Percentages are calculated using the total study sample (N = 3,540).

## Sociodemographic, lifestyle, and health-related characteristics

Table 1 presents the sociodemographic, lifestyle, and health-related characteristics of participants according to disordered eating risk. Gender was not significantly associated with disordered eating risk (45.0% males in the risk group vs. 47.1% males in the non-risk group; p = 0.256).

Academic year showed a significant association (p < 0.001). First-year students had the highest proportion of disordered eating risk (23.2% in the risk group vs. 14.7% in the non-risk group). In contrast, fourth- and fifth-year students had lower proportions in the risk group (13.8% each) compared to the non-risk group (18.9% and 17.9%, respectively).

Marital status was significantly associated with disordered eating risk in bivariate analysis (p < 0.001). Divorced participants represented 6.1% of the risk group compared to 2.4% of the non-risk group, while widowed participants represented 2.7% versus 0.9%, respectively. Smoking (20.4% vs. 17.9%, p = 0.074) and qat chewing (38.2% vs. 40.7%, p = 0.179) were not significantly associated with disordered eating risk.

Several health-related factors demonstrated strong associations. Type 1 diabetes mellitus was reported by 16.4% of students in the risk group compared to 5.1% in the non-risk group (p < 0.001). Autism spectrum disorder was reported by 16.2% of the risk group compared to 4.8% of the non-risk group (p < 0.001). A history of suicide attempts was reported by 21.9% of the risk group compared to 6.2% of the non-risk group (p < 0.001), representing more than a threefold difference. Food insecurity was reported by 38.3% of students in the risk group compared to 21.4% in the non-risk group (p < 0.001). Parental perception of being overweight was reported by 34.4% of the risk group compared to 15.0% of the non-risk group (p < 0.001).

As shown in Fig 2, most participants (76.3%, n = 2,702) reported no weight change during the six months preceding the survey. In contrast, 23.7% of participants (n = 838) reported experiencing a change in body weight during the same period.

## Psychological comorbidities and personality traits

Table 2 summarizes the distribution of psychological comorbidities and personality traits according to disordered eating risk. Most self-reported psychiatric conditions were significantly associated with disordered eating risk in bivariate

**Table 1. Sociodemographic, lifestyle, and health-related characteristics according to disordered eating risk (EAT-26 ≥ 20).**

| Variable | Category | Total n (%) | No disordered eating risk n (%) | Disordered eating risk n (%) | p-value |
|---|---|---|---|---|---|
| Gender | Male | 1644 (46.4) | 1160 (47.1) | 484 (45.0) | 0.256 |
| | Female | 1896 (53.6) | 1304 (52.9) | 592 (55.0) | |
| Academic year | First | 611 (17.3) | 361 (14.7) | 250 (23.2) | <0.001 |
| | Second | 578 (16.3) | 384 (15.6) | 194 (18.0) | |
| | Third | 589 (16.6) | 407 (16.5) | 182 (16.9) | |
| | Fourth | 614 (17.3) | 465 (18.9) | 149 (13.8) | |
| | Fifth | 588 (16.6) | 440 (17.9) | 148 (13.8) | |
| | Sixth | 560 (15.8) | 407 (16.5) | 153 (14.2) | |
| Marital status | Married | 546 (15.4) | 366 (14.9) | 180 (16.7) | <0.001 |
| | Single | 2818 (79.6) | 2017 (81.9) | 801 (74.4) | |
| | Divorced | 125 (3.5) | 59 (2.4) | 66 (6.1) | |
| | Widowed | 51 (1.4) | 22 (0.9) | 29 (2.7) | |
| Smoking | Yes | 661 (18.7) | 441 (17.9) | 220 (20.4) | 0.074 |
| | No | 2879 (81.3) | 2023 (82.1) | 856 (79.6) | |
| Qat chewing | Yes | 1413 (39.9) | 1002 (40.7) | 411 (38.2) | 0.179 |
| | No | 2127 (60.1) | 1462 (59.3) | 665 (61.8) | |
| Regular physical activity | Yes | 982 (27.7) | 623 (25.3) | 359 (33.4) | <0.001 |
| | No | 2558 (72.3) | 1841 (74.7) | 717 (66.6) | |
| Type 1 diabetes mellitus | Yes | 302 (8.5) | 126 (5.1) | 176 (16.4) | <0.001 |
| | No | 3238 (91.5) | 2338 (94.9) | 900 (83.6) | |
| Gastrointestinal disease | Yes | 926 (26.2) | 564 (22.9) | 362 (33.6) | <0.001 |
| | No | 2614 (73.8) | 1900 (77.1) | 714 (66.4) | |
| Autism spectrum disorder | Yes | 293 (8.3) | 119 (4.8) | 174 (16.2) | <0.001 |
| | No | 3247 (91.7) | 2345 (95.2) | 902 (83.8) | |
| Sleep disorders | Yes | 859 (24.3) | 499 (20.3) | 360 (33.5) | <0.001 |
| | No | 2681 (75.7) | 1965 (79.7) | 716 (66.5) | |
| Parental perception of overweight | Yes | 740 (20.9) | 370 (15.0) | 370 (34.4) | <0.001 |
| | No | 2800 (79.1) | 2094 (85.0) | 706 (65.6) | |
| Parental pressure to eat | Yes | 1803 (50.9) | 1249 (50.7) | 554 (51.5) | 0.688 |
| | No | 1737 (49.1) | 1215 (49.3) | 522 (48.5) | |
| Food insecurity | Yes | 938 (26.5) | 526 (21.4) | 412 (38.3) | <0.001 |
| | No | 2602 (73.5) | 1938 (78.6) | 664 (61.7) | |
| Past suicide attempts | Yes | 388 (11.0) | 152 (6.2) | 236 (21.9) | <0.001 |
| | No | 3152 (89.0) | 2312 (93.8) | 840 (78.1) | |
| Weight satisfaction | Satisfied | 1904 (53.8) | 1377 (55.9) | 527 (49.0) | < 0.001 |
| | Dissatisfied | 1636 (46.2) | 1087 (44.1) | 549 (51.0) | |

Values are presented as n (column percentages). Chi-square test was used. p < 0.05 considered statistically significant.

analyses. Self-reported post-traumatic stress disorder (PTSD) was reported by 8.3% of students in the risk group compared to 3.0% in the non-risk group (p < 0.001). Obsessive compulsive disorder (OCD) was reported by 8.6% of the risk group versus 4.1% of the non-risk group (p < 0.001). Social anxiety disorder was reported by 12.2% versus 7.1%, respectively (p < 0.001). Notably strong associations were observed for bipolar disorder and schizophrenia. Bipolar disorder was

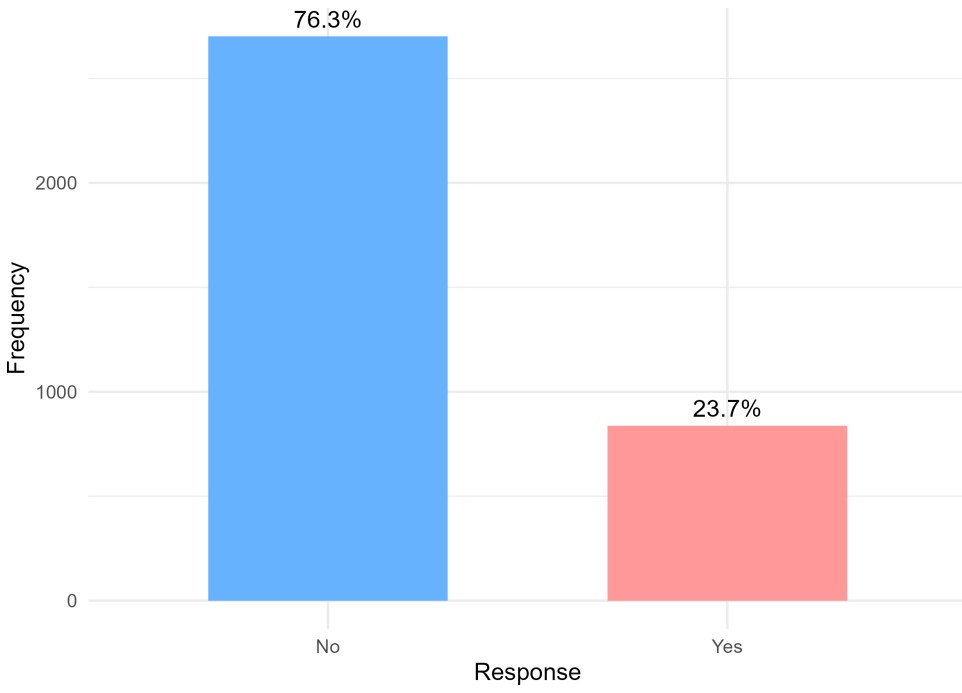

**Fig 2. Self-reported weight change among medical students during the past six months.** Distribution of participants reporting perceived body weight change within the six months preceding the survey. Weight change was self-reported and did not specify direction or magnitude. Percentages are calculated using the total study sample (N = 3,540).

reported by 3.6% of students in the risk group compared to 0.6% in the non-risk group (p < 0.001), while schizophrenia was reported by 3.0% versus 0.4%, respectively (p < 0.001), representing approximately six- to sevenfold higher proportions in the risk group.

Major depressive disorder was reported by 5.4% of the risk group compared to 2.2% of the non-risk group (p < 0.001). Borderline personality disorder was not significantly associated with disordered eating risk (1.4% vs. 0.6%, p = 0.085). Regarding personality traits, perfectionism was reported by 3.0% of students in the risk group compared to 1.5% in the non-risk group (p = 0.005), and impulsiveness was reported by 2.4% versus 1.4%, respectively (p = 0.028). Obsessive traits were not significantly associated with disordered eating risk (1.6% vs. 1.7%, p = 0.726).

## Self-reported disordered eating symptoms, frequency, and severity

Among the 1,117 participants who reported disordered eating symptoms, the most frequently reported symptom patterns were classified as other disordered eating symptoms (38.6%), followed by anorexia nervosa symptoms (27.1%) and binge eating disorder symptoms (14.1%) (Table 3). Regarding symptom frequency, 17.4% of participants reported experiencing symptoms daily, while 24.0% reported weekly symptoms. In terms of severity, 44.4% described their symptoms as mild, 42.6% as moderate, and 13.0% as severe or very severe.

## Disordered eating related behaviors

Table 4 presents the frequency of specific disordered eating–related behaviors in the full study sample. Eating binges accompanied by a perceived loss of control were reported by 37.3% of participants at least once, while 9.8% reported

**Table 2. Psychological comorbidities and personality traits according to disordered eating risk (EAT-26 ≥ 20).**

| Variable | Category | Total n (%) | No disordered eating risk n (%) | Disordered eating risk n (%) | p-value |
|---|---|---|---|---|---|
| PTSD | Yes | 164 (4.6) | 75 (3.0) | 89 (8.3) | <0.001 |
| | No | 3376 (95.4) | 2389 (97.0) | 987 (91.7) | |
| OCD | Yes | 192 (5.4) | 100 (4.1) | 92 (8.6) | <0.001 |
| | No | 3348 (94.6) | 2364 (95.9) | 984 (91.4) | |
| Social anxiety disorder | Yes | 307 (8.7) | 176 (7.1) | 131 (12.2) | <0.001 |
| | No | 3233 (91.3) | 2288 (92.9) | 945 (87.8) | |
| Bipolar disorder | Yes | 55 (1.6) | 16 (0.6) | 39 (3.6) | <0.001 |
| | No | 3485 (98.4) | 2448 (99.4) | 1037 (96.4) | |
| Schizophrenia | Yes | 42 (1.2) | 10 (0.4) | 32 (3.0) | <0.001 |
| | No | 3498 (98.8) | 2454 (99.6) | 1044 (97.0) | |
| Major depressive disorder | Yes | 112 (3.2) | 54 (2.2) | 58 (5.4) | <0.001 |
| | No | 3428 (96.8) | 2410 (97.8) | 1018 (94.6) | |
| Borderline personality disorder | Yes | 29 (0.8) | 14 (0.6) | 15 (1.4) | 0.085 |
| | No | 3511 (99.2) | 2450 (99.4) | 1061 (98.6) | |
| Perfectionism | Yes | 70 (2.0) | 38 (1.5) | 32 (3.0) | 0.005 |
| | No | 3470 (98.0) | 2426 (98.5) | 1044 (97.0) | |
| Obsessive traits | Yes | 60 (1.7) | 43 (1.7) | 17 (1.6) | 0.726 |
| | No | 3480 (98.3) | 2421 (98.3) | 1059 (98.4) | |
| Impulsiveness | Yes | 60 (1.7) | 34 (1.4) | 26 (2.4) | 0.028 |
| | No | 3480 (98.3) | 2430 (98.6) | 1050 (97.6) | |

Abbreviations: PTSD = post-traumatic stress disorder; OCD = obsessive compulsive disorder.

such episodes on a weekly basis or more frequently. Self-induced vomiting for weight control was reported by 25.1% of participants at least once. Similarly, 25.0% reported using laxatives, diet pills, or diuretics for weight control. Excessive exercise for the purpose of weight control was reported by 37.3% of participants at least once, with 11.3% engaging in this behavior on a weekly basis or more frequently.

## Multivariable predictors of disordered eating risk

Results of the multivariable logistic regression analysis are presented in Table 5. After adjusting for potential confounders, several factors remained independently associated with disordered eating risk. Regular physical activity was associated with increased odds of disordered eating risk (adjusted OR = 1.24, p = 0.017). Participants with a history of type 1 diabetes mellitus also had higher odds of disordered eating risk (adjusted OR = 1.40, p = 0.025). Parental perception of the participant being overweight was one of the strongest predictors, doubling the odds of disordered eating risk (adjusted OR = 2.02, p < 0.001). Food insecurity (adjusted OR = 1.43, p < 0.001) and a history of suicide attempts (adjusted OR = 1.94, p < 0.001) were also significant predictors. Weight satisfaction was independently protective, with satisfied participants showing lower odds of disordered eating risk (adjusted OR = 0.84, p = 0.025). Several psychological comorbidities remained strongly associated with disordered eating risk, including post-traumatic stress disorder, obsessive compulsive disorder, social anxiety disorder, major depressive disorder, bipolar disorder, and schizophrenia. Borderline personality disorder did not reach statistical significance in the adjusted model.

**Table 3. Self-reported disordered eating symptoms, frequency, and severity among participants reporting symptoms (n = 1,117).**

| Variable | Frequency (n) | Percentage (%) |
|---|---|---|
| **Type of disordered eating symptoms** | | |
| Anorexia nervosa | 303 | 8.6% |
| Bulimia nervosa | 86 | 2.4% |
| Binge Eating disorder | 157 | 4.4% |
| Avoidant/restrictive intake disorder | 140 | 4.0% |
| Other | 431 | 12.2% |
| **Frequency of symptoms** | | |
| Daily | 194 | 5.5% |
| Weekly | 268 | 7.6% |
| Monthly | 232 | 6.6% |
| Yearly | 83 | 2.3% |
| Rarely | 227 | 6.4% |
| Never | 113 | 3.2% |
| **Self-reported severity** | | |
| Mild | 496 | 14.0% |
| Moderate | 476 | 13.4% |
| Severe | 118 | 3.3% |
| Very severe | 27 | 0.8% |

Percentages are calculated among participants reporting disordered eating symptoms (n = 1,117).

**Table 4. Frequency of disordered eating related behaviors among participants (N = 3,540).**

| Behavioral question | Never n (%) | Once a month or less n (%) | 2–3 times a month n (%) | Once a week n (%) | 2–6 times a week n (%) | Once a day or more n (%) |
|---|---|---|---|---|---|---|
| Eating binges with loss of control | 2219 (62.7) | 633 (17.9) | 343 (9.7) | 106 (3.0) | 94 (2.7) | 145 (4.1) |
| Self-induced vomiting for weight control | 2650 (74.9) | 410 (11.6) | 266 (7.5) | 62 (1.8) | 65 (1.8) | 87 (2.5) |
| Use of laxatives, diet pills, or diuretics | 2653 (75.0) | 371 (10.5) | 274 (7.7) | 65 (1.8) | 77 (2.2) | 100 (2.8) |
| Excessive exercise for weight control | 2220 (62.7) | 536 (15.1) | 384 (10.8) | 114 (3.2) | 149 (4.2) | 137 (3.9) |

Percentages are calculated based on the total study sample (N = 3,540).

## Discussion

In this national multicenter cross-sectional study of Yemeni medical students, nearly one-third of participants screened positive for disordered eating risk, with 30.4% exceeding the EAT-26 cutoff score. This prevalence appears higher than estimates reported for the general population and aligns with evidence suggesting that medical students may represent a particularly vulnerable group for disordered eating behaviors. Disordered eating risk was associated with multiple socio-demographic, lifestyle, medical, and psychological factors, including regular physical activity, a history of type 1 diabetes mellitus, food insecurity, weight dissatisfaction, parental perception of being overweight, and past suicide attempts. Significant associations were also observed with several self-reported psychiatric conditions, including post-traumatic stress disorder, obsessive compulsive disorder, bipolar disorder, schizophrenia, and major depressive disorder.

In addition to elevated screening scores, a notable proportion of students reported disordered eating–related behaviors, such as binge eating, self-induced vomiting, use of weight control substances, and excessive exercise for weight control.

Table 5. Multivariable logistic regression analysis of factors associated with disordered eating risk (EAT-26 ≥ 20).

| Predictor | Adjusted OR | 95% CI | p-value |
|---|---|---|---|
| **Regular physical activity** | 1.24 | 1.04 to 1.47 | 0.017 |
| **Type 1 diabetes mellitus** | 1.40 | 1.04 to 1.89 | 0.025 |
| **Gastrointestinal disease** | 1.06 | 0.89 to 1.27 | 0.516 |
| **Autism spectrum disorder** | 1.33 | 0.99 to 1.80 | 0.061 |
| **Sleep disorders** | 1.12 | 0.93 to 1.35 | 0.233 |
| **Parental perception of overweight** | 2.02 | 1.68 to 2.44 | < 0.001 |
| **Food insecurity** | 1.43 | 1.20 to 1.70 | < 0.001 |
| **Past suicide attempts** | 1.94 | 1.50 to 2.52 | < 0.001 |
| **Weight satisfaction** | 0.84 | 0.71 to 0.98 | 0.025 |
| **Post-traumatic stress disorder** | 2.20 | 1.54 to 3.14 | < 0.001 |
| **Obsessive compulsive disorder** | 2.22 | 1.61 to 3.07 | < 0.001 |
| **Social anxiety disorder** | 2.11 | 1.63 to 2.73 | < 0.001 |
| **Bipolar disorder** | 3.52 | 1.86 to 6.68 | < 0.001 |
| **Schizophrenia** | 5.96 | 2.81 to 12.68 | < 0.001 |
| **Major depressive disorder** | 2.66 | 1.77 to 4.00 | < 0.001 |
| **Borderline personality disorder** | 1.98 | 0.91 to 4.30 | 0.085 |
| **Perfectionism** | 2.01 | 1.21 to 3.35 | 0.007 |
| **Impulsiveness** | 2.65 | 1.55 to 4.54 | < 0.001 |

OR = odds ratio; CI = confidence interval. Variables were entered into the model based on p < 0.20 in bivariate analysis. Model fit was assessed using the Hosmer-Lemeshow test. Statistical significance was set at p < 0.05.

Although marital status showed a significant association in bivariate analysis, it was not retained as an independent predictor in the adjusted model. Similarly, smoking, qat chewing, and parental pressure to eat were not independently associated with disordered eating risk.

The finding that 30.4% of medical students in this study screened positive for disordered eating risk exceeds pooled prevalence estimates reported in recent meta-analyses of medical students, including 17.35% by Fekih-Romdhane et al. (2022) and 15.1% by Jahrami et al. (2024) [25,26]. It is important to note that these estimates are largely derived from screening-based studies using instruments comparable to the EAT-26, rather than diagnostic interviews, which supports the appropriateness of comparison. Our findings are also consistent with recent data from the Middle East and North Africa region, where Belhaj Salem et al. (2025) reported that 24.8% of medical students were at risk of disordered eating, with prevalence estimates ranging from 32% to 35% in countries such as Egypt, Saudi Arabia, and the United Arab Emirates [22].

Although female participants in our study demonstrated a higher proportion of disordered eating risk compared to males, this difference was not statistically significant. This finding contrasts with traditional assumptions that disordered eating disproportionately affects females but aligns with evidence from recent meta-analyses reporting no significant gender differences among medical students [26]. Additionally, emerging literature increasingly recognizes that a substantial proportion of males experience disordered eating behaviors, particularly in high-stress academic environments [27]. Therefore, our findings support the need for gender-inclusive screening and intervention strategies. The statement comparing prevalence in medical trainees to the general population should be interpreted cautiously, as differences in study design and screening tools limit direct comparison [28].

Medical students are exposed to a unique combination of academic, psychological, and sociocultural stressors that may increase vulnerability to disordered eating behaviors. The demanding nature of medical training, characterized by

 

academic overload, competitive environments, high expectations, and exposure to illness and mortality, contributes to sustained psychological stress [25,29,30]. This stress frequently co-occurs with depression, anxiety, burnout, and perfectionistic traits, all of which were significantly associated with disordered eating risk in the present study [25,30]. When adaptive coping strategies are limited, maladaptive behaviors such as disordered eating may emerge as mechanisms for emotional regulation or perceived control [28,30].

Sociocultural influences, including internalization of thin-ideal body standards promoted through media and societal expectations, further contribute to body dissatisfaction among medical students [25,30] Familial factors also play an important role. In this study, parental perception of the student being overweight was one of the strongest predictors of disordered eating risk, consistent with prior research highlighting the impact of weight-related criticism and stigma on eating behaviors [28,30]. Interventions targeting weight self-stigma and body image dissatisfaction have demonstrated promising results in reducing disordered eating risk, underscoring the relevance of these findings [31].

A significant association was observed between disordered eating risk and a history of type 1 diabetes mellitus, while gastrointestinal disorders did not remain independently associated after multivariable adjustment. The management of type 1 diabetes requires continuous monitoring of food intake, body weight, and insulin dosing, which may increase vulnerability to maladaptive eating behaviors, including intentional insulin restriction for weight control. This phenomenon, often referred to as diabulimia, has been reported in 20% to 60% of individuals with type 1 diabetes and is associated with severe complications and increased mortality risk [32–34]. Conversely, disordered eating behaviors such as binge eating, purging, or restrictive intake can disrupt glycemic control and complicate diabetes management [32].

Although gastrointestinal disorders were common in the study population, their association with disordered eating risk did not persist after accounting for psychological comorbidities. This finding is consistent with previous research demonstrating substantial overlap between gastrointestinal symptoms and disordered eating, often mediated by anxiety and depressive disorders [35,36] Gastrointestinal symptoms may contribute to avoidant or restrictive eating patterns driven by fear of discomfort rather than body image concerns, while disordered eating behaviors themselves can result in significant gastrointestinal sequelae, including esophageal injury, gastroparesis, constipation, and functional bowel symptoms [37]. The bidirectional relationship between these conditions likely contributes to diagnostic complexity.

Psychiatric comorbidity and suicidality were strongly associated with disordered eating risk in this study, with 11.0% of participants reporting a history of suicide attempts. This finding is consistent with extensive evidence identifying eating disorders and disordered eating behaviors as conditions associated with elevated suicide risk [38,39] Anorexia nervosa is associated with particularly high suicide mortality, while lifetime suicide attempt rates are also substantial among individuals with bulimic behaviors [39].

In line with previous research, our findings indicate that psychiatric comorbidity is common among individuals at risk of disordered eating. Significant associations were observed with post-traumatic stress disorder, obsessive compulsive disorder, social anxiety disorder, major depressive disorder, bipolar disorder, and schizophrenia. Prior studies have similarly demonstrated that mood and anxiety disorders frequently co-occur with disordered eating and contribute to greater symptom severity and poorer outcomes [40,41]. Trauma exposure and post-traumatic stress symptoms, in particular, have been identified as important contributors to the complexity and chronicity of disordered eating behaviors [41,42].

## Strengths and limitations

This study included a large sample of medical students recruited from fifteen colleges across multiple regions of Yemen and from all academic years, providing broad geographic and academic representation. The use of the EAT-26, a widely validated screening instrument with strong internal consistency in the present sample, strengthens the reliability of the screening assessment.

Several limitations should be considered when interpreting the findings. First, the cross-sectional design precludes inference regarding causality or temporal relationships between disordered eating risk and associated factors. Second,

a non-probability convenience sampling strategy was used, and participation was voluntary through online distribution channels. Because the exact number of students who received the survey invitation could not be determined, a response rate could not be calculated. This may limit assessment of representativeness and introduces the possibility of selection bias.

All variables were based on self-report, including psychiatric diagnoses, chronic medical conditions, suicide history, and anthropometric measures used to calculate body mass index. Psychiatric conditions were not clinically verified, and self-reported height and weight may be subject to reporting bias. These factors may have led to misclassification or measurement error.

Additionally, the EAT-26 is a screening instrument rather than a diagnostic tool and therefore cannot be used to estimate the prevalence of clinically confirmed eating disorders. Future longitudinal studies are warranted to clarify temporal relationships and better understand the progression and persistence of disordered eating risk among medical students. Consideration could be given to incorporating screening and mental health support strategies within medical training institutions, although further research is needed to evaluate the effectiveness of such approaches in this context.

## Conclusion

This national multicenter cross-sectional study indicates that disordered eating risk is common among medical students in Yemen, with 30.4% screening positive on the EAT-26. Disordered eating risk was associated with several sociodemographic, medical, and psychological factors, including food insecurity, parental perception of being overweight, weight dissatisfaction, type 1 diabetes mellitus, past suicide attempts, and multiple self-reported psychiatric conditions. Risk was observed in both male and female students. Given the cross-sectional design and reliance on self-reported data, causal relationships cannot be established. These findings suggest that consideration may be given to incorporating screening and supportive mental health resources within medical training institutions. Further longitudinal research is needed to clarify temporal relationships and inform preventive strategies.

## Supporting information

**S1 File. STROBE checklist for cross-sectional studies.** Checklist of items that should be included in reports of cross-sectional studies.
(PDF)

## Acknowledgments

We would like to express our deepest gratitude to the Yemeni medical student community for their invaluable support and active participation in this research. Their collaboration, commitment, and enthusiasm toward advancing medical knowledge were fundamental to the success of this study. Our heartfelt thanks go to Professor Mohammed Abdullah Al-Eryani for his generous support and steadfast encouragement throughout all stages of this research. His continuous guidance and belief in the potential of this work played a vital role in its completion and quality.

We also extend our sincere appreciation to Professor Mohammed Alshehri for his exceptional mentorship. His insightful feedback and thoughtful direction significantly enhanced the academic rigor and impact of this study. Special thanks are due to Saif Alaribi and Yusra Rashed Al-Subaihi, whose assistance during the data collection phase, though limited in scope, meaningfully contributed to the progress of this project. We are also grateful to the following individuals for their dedicated efforts in data collection:

Mohammed R. Arrabyee, Ziad Mohammed AL-Othrubi, Amjed Al-Jahmi, Hajar Aqabi, Akram Al-Wadei, Fatima Odhah, Khulood Al-Khateeb, Fatima Al-Mutawakel, Ahmed Ahmed, Yousef Al-Qadimi, and Akram Al-Wadei.

Their valuable contributions were instrumental in ensuring the thoroughness and success of the study.

## Author contributions

**Conceptualization:** Naji Al-bawah, Mohamed Baklola, Ahmed Al-Eryani.

**Data curation:** Mohamed Baklola, Anas H. Khalifeh, Ehab sharyan, Ahmed Al-Eryani, Fayda Al-wesabi, Aaida Al Wesabi, Ahdab Al-Jaberi, Ahmed Abdulmughni, Hisham Salman.

**Formal analysis:** Naji Al-bawah, Mohamed Baklola, Anas Zakarya Nourelden, Aaida Al Wesabi, Ahdab Al-Jaberi, Hisham Salman, Amira Yasmine Benmelouka.

**Funding acquisition:** Naji Al-bawah, Mohamed Baklola, Anas H. Khalifeh, Ehab Sharyan, Anas Zakarya Nourelden, Saif Alaribi, Aaida Al Wesabi, Ahdab Al-Jaberi, Ahmed Abdulmughni, Amira Yasmine Benmelouka.

**Investigation:** Naji Al-bawah, Anas H. Khalifeh, Ehab Sharyan, Saif Alaribi, Ahdab Al-Jaberi, Ghailan Al-Jarbani, Amira Yasmine Benmelouka.

**Methodology:** Naji Al-bawah, Mohamed Baklola, Ahmed Abdulmughni, Amira Yasmine Benmelouka.

**Project administration:** Sama Shamsan, Ghailan Al-Jarbani.

**Resources:** Naji Al-bawah, Sama Shamsan, Ahmed Abdulmughni, Ghailan Al-Jarbani.

**Supervision:** Mohamed Baklola, Sama Shamsan.

**Validation:** Mohamed Baklola, Sama Shamsan.

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
