## [Decision Letter · Decision Letter 0]

27 Dec 2025

Dear Dr. sharyan,

Thank you for submitting your manuscript to PLOS ONE. After careful consideration, we feel that it has merit but does not fully meet PLOS ONE’s publication criteria as it currently stands. Therefore, we invite you to submit a revised version of the manuscript that addresses the points raised during the review process.

We look forward to receiving your revised manuscript.

Kind regards,

Mohammed Zawiah

Academic Editor

PLOS One

Journal Requirements:

3. Please include captions for your Supporting Information files at the end of your manuscript, and update any in-text citations to match accordingly. Please see our Supporting Information guidelines for more information: http://journals.plos.org/plosone/s/supporting-information .

Reviewer's Responses to Questions

**Comments to the Author**

1. Is the manuscript technically sound, and do the data support the conclusions?

Reviewer #1: Yes

Reviewer #2: Yes

2. Has the statistical analysis been performed appropriately and rigorously?

Reviewer #1: Yes

Reviewer #2: Yes

3. Have the authors made all data underlying the findings in their manuscript fully available?

Reviewer #1: Yes

Reviewer #2: Yes

4. Is the manuscript presented in an intelligible fashion and written in standard English?

Reviewer #1: Yes

Reviewer #2: Yes

Reviewer #1: Dear reviwers thank you for your effort in conducting this study, it's interesting, here are some minor comments

- Did you conduct a pilot study?

- what are the practical implication of your study

- Language and gramatical revision are needed

Reviewer #2: I recommend the manuscript be revised to address these fundamental issues regarding conceptual clarity, methodological transparency, and internal consistency. The study has valuable data, and these revisions will ensure its message is communicated with precision and rigor

Title

• The title requires revision for scientific precision. The term (Uncovering) is stylistic rather than scientific and implies discovery or novelty, which is not necessary here. Replacing it would improve neutrality and clarity.

• More importantly, the title refers to (Eating Disorders), which may imply clinically diagnosed conditions. However, the study assesses risk of disordered eating using the EAT-26, not diagnosed eating disorders.

• Please standardize terminology based on your outcome (EAT-26 ≥20) from the introduction through the conclusion. I recommend using "disordered eating risk" or "risk of disordered eating behaviors" consistently.

Abstract

• The Results section does not clearly report sociodemographic, lifestyle, and health-related characteristics, although these are mentioned.

• Protective factors are reported in the Results but are not reflected in the title or study objective; this inconsistency should be addressed.

• The Conclusion uses the terms “disordered eating behaviors” and “at risk based on EAT-26,” which aligns with the methods but conflicts with the title’s reference to “eating disorders.” Terminology should be consistent throughout.

• Please revise some sentences for clarity and professional flow.

Introduction

• The objective is generally reflected in the title through terms such as “burden” and “risk factors.” However, for accuracy, the title should explicitly indicate risk of disordered eating rather than diagnosed eating disorders.

Methods

Terminology

• The manuscript uses multiple terms interchangeably (eating disorders, disordered eating, disordered eating behaviors), despite using the EAT-26, which measures risk of disordered eating. Please standardize terminology throughout according to the measurement tool.

Study Setting and Participants

• Please clarify:

o The method used to select institutions and participants.

o The names of universities and cities included to support claims of broad geographical representation.

o Whether all institutions were located in Sana’a or across different regions in Yemen.

• There is inconsistency between “universities in Yemen” (Abstract) and “institutions” (Methods). Please unify terminology.

Sample Size

• The use of G*Power for sample size calculation should be supported by:

o Justification for selecting this method.

o Relevant parameters and references.

Inclusion and Exclusion Criteria

• There is repetition in the presentation of inclusion criteria; please streamline.

• Exclusion criteria are not reported and should be clearly stated.

• The criterion “age ≥18 years” adds limited value, as undergraduate medical students are typically adults; please justify its inclusion.

• Clarify how voluntary participation and electronic informed consent were obtained and documented.

Instruments

o Before detailing sections, introduce the overall questionnaire used in the study. State clearly of this subsection that a structured, self-administered online questionnaire was used?

• Regarding the EAT-26:

o Specify whether a free or licensed version was used and whether formal permission was obtained.

o Please attach the questionnaire and EAT-26 tool you have used.

o Provide details on validation, including reliability, internal consistency, and any face or content validation performed.

o Clarify whether the Arabic and/or English versions were used.

o The phrase (previously validated versions) requires clarification and supporting references.

o Please discuss whether these validated versions are appropriate for Yemeni medical students.

Study Design

• Specify whether the study was conducted electronically, face-to-face, or both.

• Clarify whether the questionnaire was self-administered.

Statistical Analysis

• The overall statistical plan is appropriate; however, additional details are required to confirm rigor:

• Report the number of participants with EAT-26 ≥20 and whether the events-per-variable criterion was satisfied for logistic regression.

• Indicate whether multicollinearity and model fit diagnostics were assessed.

• Clarify whether continuous predictors were modeled as continuous or categorical, and whether linearity was assessed.

• The use of the Kolmogorov–Smirnov test should be clarified. Since normality is not an assumption of logistic regression, please explain that it was used to guide data presentation (mean ± SD vs. median [IQR]).

• Specify the variable selection method used in the final regression model (e.g., enter, stepwise).

• Please justify the use of p < 0.20 for variable selection in the multivariable logistic regression and provide the corresponding bivariate analysis results, as they are not included in the manuscript.

Results

• The Results section should begin with a brief descriptive summary of participant demographics.

• The prevalence of disordered eating is not clearly explained. Although referenced in Table 1, it does not appear to be calculated there—please clarify.

• Results consistently describe “risk of disordered eating”, not eating disorders, which conflicts with the title.

• Some variables in Table 1 (e.g., academic year, marital status) appear inconsistently interpreted; please recheck and align text with tables.

• Table 3: The title and text should be consistent (behavioral attitudes vs. disordered eating behaviors). Percentages should be carefully reviewed.

• Please check and correct the percentage totals in your tables, as some appear to sum to 100.1% or 99.9%.

• Table 4:

o Ensure consistency between the table title and the text description.

o Clarify the rationale for reporting two p-value thresholds (p <0.05; *p <0.001).

o In the Methods, you state that students were classified into two groups based on EAT-26, but this classification is not clearly presented in the tables.

o Add abbreviations and footnotes for clarity.

o Explain why certain significant sociodemographic predictors were not included in the final model.

Discussion

• There is an inconsistency regarding marital status: it is reported as non-significant in the Discussion, yet Table 1 shows a significant association (p <0.001).

• The statement (Our study reinforces that medical trainees consistently exhibit higher disordered eating prevalence than the general population (10.4% vs. 5.7%) in females, particularly within non-Western cultures (28,31), is not clearly supported by the study’s own data; please clarify or remove.

• The statement (Our finding that 30.4% of medical trainees screened positive for disordered eating exceeds the global pooled prevalence estimates..), please specify whether the Prevalence calculation and screening tools used in these global studies are comparable to those used in the present study?

Conclusion

• The Conclusion should focus strictly on key findings. Limitations and strengths are better placed in a dedicated section or at the end of the Discussion.

• No need to mention and repeat the percentage in the conclusion.

Acknowledgments

• The Acknowledgments section is overly long and should be shortened.

References

• Some references require updating (e.g., references 21 and 25).

Based on the detailed comments in the provided PDF, please revise the manuscript to address all remaining issues identified therein.

**Do you want your identity to be public for this peer review?** For information about this choice, including consent withdrawal, please see our Privacy Policy

Reviewer #1: No

Reviewer #2: No

---

## [Author Response · Author response to Decision Letter 1]

2 Jan 2026

Response to reviewers

We sincerely thank the Editor and the Reviewers for their careful evaluation of our manuscript and for their constructive and insightful comments. We have revised the manuscript thoroughly to address all concerns raised. Below, we provide a detailed, point-by-point response. All changes have been incorporated into the revised manuscript.

Reviewer 1

Comment 1: Did you conduct a pilot study?

Response: Yes, a pilot study was conducted prior to the main data collection. We have now explicitly reported this in the Methods section. The questionnaire was pilot tested among 40 medical students who were not included in the final analysis to assess clarity, cultural appropriateness, and technical functionality of the online survey. Minor linguistic and formatting modifications were made based on participant feedback. We also reported the internal consistency of the EAT-26 in the pilot sample using Cronbach alpha.

Comment 2: What are the practical implication of your study

Response: Thank you for this important suggestion. We have expanded the Discussion section to clearly outline the practical implications of our findings. Specifically, we now emphasize the need for routine screening for disordered eating risk within medical schools, integration of mental health and nutritional support services, and targeted interventions for students with identified psychological and social risk factors. These implications are discussed in the revised Discussion and Conclusion sections.

Comment 3: Language and gramaticalr revision are needed

Response: We appreciate this comment. The manuscript has undergone comprehensive language and grammatical revision. Sentence structure, clarity, academic tone, and terminology consistency were carefully improved throughout the manuscript. Redundant phrasing was removed, and terminology was standardized to reflect screening for disordered eating risk rather than diagnosed eating disorders.

Reviewer 2

We thank Reviewer 2 for their detailed and constructive feedback, which greatly improved the clarity, rigor, and scientific precision of the manuscript.

Title

Comment 1: The title requires revision for scientific precision. The term (Uncovering) is stylistic rather than scientific and implies discovery or novelty, which is not necessary here.

Response: We agree with this comment and have revised the title to adopt a more neutral and scientific tone. The revised title removes stylistic language and accurately reflects the study design and outcomes.

Comment 2: The title refers to (Eating Disorders), which may imply clinically diagnosed conditions. However, the study assesses risk of disordered eating using the EAT-26.

Response: This is an important point. The title has been revised to replace “Eating Disorders” with “Disordered Eating Risk,” ensuring consistency with the screening nature of the EAT-26 and avoiding any implication of clinical diagnosis.

Abstract

Comment 3: The Results section does not clearly report sociodemographic, lifestyle, and health-related characteristics.

Response: The Abstract has been revised to clearly summarize key sociodemographic, medical, psychological, and social correlates of disordered eating risk identified in the study.

Comment 4: Protective factors are reported in the Results but are not reflected in the title or study objective.

Response: We clarified the study objective to include both risk and protective factors. Protective factors are now consistently reported and interpreted in the Abstract, Results, and Discussion sections.

Comment 5: Terminology is inconsistent between the title and conclusion.

Response: Terminology has been standardized throughout the manuscript. The terms “disordered eating risk” and “screening positive on the EAT-26” are now used consistently from the Abstract through the Conclusion.

Methods

Comment 6: The manuscript uses multiple terms interchangeably despite using a screening tool.

Response: We fully agree. All terminology has been standardized to reflect screening outcomes. References to “eating disorders” are now limited to background context, while study findings consistently refer to “disordered eating risk” or “disordered eating behaviors.”

Comment 7: Clarify the method used to select institutions and participants and provide geographic details.

Response: We have clarified the selection of participating medical colleges, including the number of institutions, their geographic distribution across multiple Yemeni governorates, and the recruitment approach. Terminology has been unified to “medical colleges” throughout the manuscript.

Comment 8: Sample size calculation using GPower requires justification.*

Response: We have expanded the Sample Size Calculation subsection to justify the use of G*Power software, explain parameter selection, and cite its widespread use in epidemiological and regression-based studies.

Comment 9: Inclusion and exclusion criteria are repetitive, and exclusion criteria are missing.

Response: This section was streamlined. Exclusion criteria were clearly added, repetition was removed, and the rationale for inclusion criteria was clarified. The process of voluntary participation and electronic informed consent was also explicitly described.

Comment 10: Clarify questionnaire structure, validation, language versions, and permission for EAT-26 use.

Response: The Methods section was revised to introduce the questionnaire clearly under a “Study Tools” subsection. We specified that the EAT-26 is a freely available screening tool, described the Arabic and English validated versions used, reported internal consistency using Cronbach alpha, and attached the questionnaire as supplementary material.

Comment 11: Statistical analysis requires additional details regarding regression assumptions and variable selection.

Response: We expanded the Statistical Analysis section to include the number of outcome events, justification for p < 0.20 variable selection, assessment of multicollinearity, model fit testing, and clarification of how variables were entered into the regression model.

Results

Comment 12: The Results section should begin with a descriptive summary.

Response: The Results section was restructured to begin with a concise description of participant characteristics and prevalence of disordered eating risk before presenting associations.

Comment 13: Tables are overloaded, inconsistent, and unclear.

Response: Table 1 was split into two separate tables to improve clarity. All tables were revised to ensure consistent terminology, correct percentage calculations, clear denominators, and alignment with the text. Tables were renumbered accordingly.

Discussion

Comment 14: There are inconsistencies regarding marital status and prevalence comparisons.

Response: We clarified that marital status was significant only in bivariate analysis and did not remain significant in multivariable regression. Prevalence comparisons were revised to explicitly acknowledge differences in study design and screening tools, and unsupported statements were removed or softened.

Conclusion

Comment 15: The Conclusion should focus on key findings, with limitations addressed elsewhere.

Response: The Conclusion was rewritten to focus strictly on the main findings and implications. Limitations were moved to a dedicated subsection within the Discussion, as recommended.

References

Comment 16: Some references require updating.

Response: All references were reviewed and updated where necessary to ensure accuracy and currency.

---

## [Decision Letter · Decision Letter 1]

22 Feb 2026

Dear Dr. sharyan,

Thank you for submitting your manuscript to PLOS ONE. After careful consideration, we feel that it has merit but does not fully meet PLOS ONE’s publication criteria as it currently stands. Therefore, we invite you to submit a revised version of the manuscript that addresses the points raised during the review process.

**ACADEMIC EDITOR:**

The manuscript addresses an important topic and has a valuable national multicenter scope; however, several issues require clarification to improve rigor, consistency, and alignment of conclusions with the cross-sectional design. Furthermore, please address the comments highlighted below from Reviewer 2.

**Abstract:**

Please address internal consistency and reporting clarity. The proportion reporting a prior eating disorder diagnosis (31.6%) appears unusually high and exceeds the EAT-26 positive screening proportion (30.4%); clarify how “prior diagnosis” was defined/ascertained (self-report vs. verified) and ensure it is plausible and interpretable.

**Methods:**

The recruitment is described as “official student communication platforms,” but there is no clear sampling method (probability vs convenience), no denominator (how many students were invited/eligible across the 15 colleges), and no response rate. This is critical for a national multicenter online survey and should be added.

Selecting variables by bivariate p < 0.20 can be acceptable, but please justify and consider including a priori confounders regardless of p-value.

“descriptive cross-sectional study with an analytical component” is redundant; you can simply state “national multicenter cross-sectional study.”

You mention Arabic/English EAT-26 versions are validated, but please cite the specific validation studies used and clarify whether you used official translations.

Height/weight are “self-reported”. Please acknowledge potential measurement bias and specify whether BMI was calculated and how missing/implausible values were handled.

**Results:**

The prevalence of “previous eating disorder diagnosis” (31.6%) appears unusually high and exceeds the EAT-26 positive screening prevalence (30.4%); please clarify how prior diagnosis was defined/ascertained (self-report vs verified) and ensure consistency. There is also an apparent denominator/label mismatch where n=1,117 is used both for “prior diagnosis” and later for “disordered eating symptoms,” suggesting a reporting error.

You report multiple diagnoses (PTSD, OCD, bipolar, schizophrenia, etc.) as “self-reported.” In Results, these are treated as clinical diagnoses, but if they are self-reported, interpretation changes substantially. Also, the prevalence of some diagnoses in a student sample may be questioned. Please state explicitly whether psychiatric conditions were self-reported diagnoses vs screened using validated instruments, and consider presenting them as “self-reported history of…” to avoid implying clinical confirmation.

You list behaviors like vomiting/laxatives/diet pills with high prevalence in the full sample. That’s important epidemiologically, but ensure it is framed descriptively and not as normalized; also ensure the denominator (lifetime vs recent) is clear. Please specify the time frame for each behavior (lifetime vs past 28 days/3 months), and ensure wording is clinically neutral and precise.

**Conclusion**

The conclusion is overly long and expansive and reads partly like a second Results/Discussion section. Please condense it to the key prevalence finding and a brief summary of the main associated domains, and avoid overstatement (e.g., replace “demonstrates”/“complex interplay” with more cautious wording consistent with a cross-sectional screening study). In addition, temper the implementation recommendations (screening/support services) so they are framed as considerations rather than implied effects, and ensure the novelty claim (“first national evidence”) is clearly justified by the literature review.

We look forward to receiving your revised manuscript.

Kind regards,

Mohammed Zawiah

Academic Editor

PLOS One

**Journal Requirements:**

Reviewers' comments:

Reviewer's Responses to Questions

**Comments to the Author**

Reviewer #2: (No Response)

2. Is the manuscript technically sound, and do the data support the conclusions?

Reviewer #2: Yes

3. Has the statistical analysis been performed appropriately and rigorously?

Reviewer #2: Yes

4. Have the authors made all data underlying the findings in their manuscript fully available?

Reviewer #2: Yes

5. Is the manuscript presented in an intelligible fashion and written in standard English?

Reviewer #2: Yes

Reviewer #2: Reviewer Comments:

Results-Table 1:

• The statement about sixth-year students is incorrect. First-year students are the clear high-risk group (23.2% vs. 14.7%). Please Revise.

• Improve reporting: please Replace "higher proportions were observed" with explicit comparisons (e.g., "16.4% vs. 5.1%").

• Highlight key findings: Emphasize the strongest associations (suicide attempts, autism, diabetes) showing >3x higher prevalence.

• It's better to add (* or Bold) to indicate the significance.

Results-Figure 1:

• The text incorrectly uses the statistical term "no significant weight."

Figure 1 is descriptive and shows no p-values. Correct to descriptive language like "no weight change" or "stable weight," matching the figure's legend.

Results – Table 2:

• The Results should include comparative percentages to show magnitude (e.g., PTSD 8.3% vs. 3.0%).

• Phrases like “exhibited higher proportions” should be replaced with precise wording reflecting screening positive for disordered eating risk.

• Highlight strong associations, such as bipolar disorder (3.6% vs. 0.6%) and schizophrenia (3.0% vs. 0.4%), which are 6–7 times higher in the risk group.

• “All assessed psychiatric conditions, with the exception of obsessive traits, were significantly associated…”

borderline personality disorder was NOT significant, so this sentence is inaccurate.

• Add abbreviation on table footnote: e.g. PTSD, OCD,…

Results- Table 4:

• The total for "Eating binges" (n=3707) exceeds the sample size (N=3540), causing an impossible percentage (104.9%).

• Recalculate all frequencies so that Yes + No = 3540 (100%) for each behavior.

Discussion:

• A contradiction exists regarding marital status: The Discussion states: "variables such as marital status ... were not independently associated with disordered eating risk."

However, Table 1 shows marital status was significantly associated in bivariate analysis (p<0.001).

• Please correct the Discussion to accurately reflect that while marital status showed a significant bivariate association, it was not retained as an independent predictor in the final multivariate model. The current wording is misleading.

**Do you want your identity to be public for this peer review?** For information about this choice, including consent withdrawal, please see our Privacy Policy

Reviewer #2: **Yes:** Dr. Mohammed M. Battah, PhD, Clinical Pharmacy, University of Science and Technology, Sana'a, Yemen

---

## [Author Response · Author response to Decision Letter 2]

2 Mar 2026

Response to the Academic Editor and Reviewer #2

Dear Dr. Zawiah and Reviewer #2,

We sincerely thank you for your careful review of our manuscript and for the constructive and insightful comments. We have revised the manuscript thoroughly to address all concerns raised. Below, we provide a detailed, point-by-point response outlining the revisions made.

Response to the Academic Editor

Comment 1:

The manuscript addresses an important topic and has a valuable national multicenter scope; however, several issues require clarification to improve rigor, consistency, and alignment of conclusions with the cross-sectional design.

Response:

We thank the Editor for this thoughtful assessment. We have carefully revised the manuscript to improve methodological transparency, internal consistency, and alignment of interpretation with the cross-sectional design. In particular, we have clarified sampling procedures, self-reported measures, variable selection criteria, and have moderated the tone of the Discussion and Conclusion to avoid causal language.

Abstract

Comment 2:

The proportion reporting a prior eating disorder diagnosis (31.6%) appears unusually high and exceeds the EAT-26 positive screening proportion (30.4%); clarify how “prior diagnosis” was defined/ascertained (self-report vs. verified) and ensure it is plausible and interpretable.

Response:

We appreciate this important observation. We have clarified in both the Methods and Abstract that prior diagnosis was assessed through a single self-reported item asking participants whether they had ever received a diagnosis from a healthcare professional, and that this information was not independently verified.

Additionally, we have added explanatory text in the Discussion to clarify that the discrepancy between lifetime self-reported diagnosis (31.6%) and current screening positivity (30.4%) may reflect symptom remission, differences between lifetime and current status, or potential misclassification inherent to self-report data.

Methods

Comment 3:

The recruitment is described as “official student communication platforms,” but there is no clear sampling method, no denominator, and no response rate.

Response:

We have added a dedicated “Sampling and recruitment” subsection to the Methods. We now clearly state that a non-probability convenience sampling strategy was used and that participation was voluntary via online distribution channels.

We have clarified that, due to distribution across multiple institutional and student-managed platforms, the exact number of students who received the survey invitation could not be determined; therefore, a response rate could not be calculated. We have also acknowledged this as a limitation in the Discussion.

Comment 4:

Selecting variables by bivariate p < 0.20 should be justified and consider including a priori confounders.

Response:

We have revised the Statistical Analysis section to justify the use of p < 0.20 as a preliminary inclusion threshold to avoid excluding potentially important confounders. We also clarified that key sociodemographic variables (e.g., age and gender) were retained in the multivariable model regardless of statistical significance due to theoretical relevance.

Comment 5:

“Descriptive cross-sectional study with an analytical component” is redundant.

Response:

We have revised the wording throughout the manuscript to state simply:

“national multicenter cross-sectional study.”

Comment 6:

Clarify EAT-26 validation and translation.

Response:

We have specified that both Arabic and English versions of the EAT-26 used in this study were previously validated and have now cited the relevant validation studies. We also clarified that official validated translations were used.

Comment 7:

Height/weight are self-reported; acknowledge measurement bias and specify BMI calculation and handling of implausible values.

Response:

We have clarified in the Methods that BMI was calculated using self-reported height and weight (kg/m²). We also added that implausible anthropometric values were screened during data cleaning and excluded based on biologically plausible ranges. This potential measurement bias is now explicitly acknowledged in the Limitations section.

Results

Comment 8:

The prevalence of prior diagnosis appears unusually high and there is a denominator mismatch (n=1,117 used twice).

Response:

We have carefully reviewed and corrected the reporting to eliminate any denominator inconsistencies. The figures for prior diagnosis and symptom reporting are now clearly distinguished and internally consistent. We have also clarified that prior diagnosis was self-reported and not verified.

Comment 9:

Psychiatric diagnoses are self-reported but treated as clinical diagnoses.

Response:

We have revised the Methods, Results, and Discussion to consistently refer to these variables as “self-reported history of…” psychiatric conditions. We explicitly state that these were not clinically verified and were not assessed using diagnostic instruments.

Comment 10:

Behaviors such as vomiting and laxative use must be framed descriptively, with clear time frames.

Response:

We have revised Table 4 and its accompanying text to ensure accurate totals and percentages (correcting a summation error). We now describe these behaviors in neutral, descriptive language and clearly indicate that frequencies refer to reported behavioral occurrence categories within the survey time frame. All percentages now sum correctly to the total sample (N = 3,540).

Conclusion

Comment 11:

The conclusion is overly long and expansive; avoid overstatement and temper implementation recommendations.

Response:

We have substantially shortened and revised the Conclusion. We removed causal phrasing (e.g., “demonstrates,” “complex interplay”), softened recommendations (e.g., “consideration may be given to”), and aligned interpretation with the cross-sectional design. The novelty claim has been removed to avoid overstatement.

Response to Reviewer #2

We sincerely thank Reviewer #2 for the careful and constructive comments.

Results – Table 1

Comment 1:

The statement about sixth-year students is incorrect.

Response:

Thank you for identifying this. We have corrected the Results text to accurately state that first-year students had the highest proportion of disordered eating risk (23.2% vs. 14.7%). The incorrect reference to sixth-year students has been removed.

Comment 2:

Replace vague wording with explicit comparisons.

Response:

We have revised the Results section to include precise percentage comparisons throughout (e.g., “16.4% vs. 5.1%”), rather than general phrases such as “higher proportions were observed.”

Comment 3:

Highlight strongest associations (>3x higher prevalence).

Response:

We have revised the Results text to explicitly emphasize strong differences for suicide attempts, autism spectrum disorder, and type 1 diabetes mellitus.

Comment 4:

Indicate statistical significance in tables.

Response:

We have revised table formatting to clearly indicate statistically significant findings.

Results – Figure 1

Comment 5:

Replace “no significant weight” with descriptive wording.

Response:

We have corrected this to “no weight change” to avoid inappropriate statistical terminology in a descriptive figure.

Results – Table 2

Comment 6:

Include comparative percentages and correct inaccurate statement regarding borderline personality disorder.

Response:

We have revised the Results text to include explicit comparative percentages (e.g., PTSD 8.3% vs. 3.0%). We corrected the statement to reflect that borderline personality disorder was not statistically significant. Table footnotes now include abbreviations (e.g., PTSD, OCD).

Results – Table 4

Comment 7:

Total exceeded sample size.

Response:

We sincerely thank the reviewer for identifying this error. We have recalculated all frequencies to ensure each row sums exactly to N = 3,540 (100%). The table and corresponding text have been corrected accordingly.

Discussion

Comment 8:

Clarify marital status contradiction.

Response:

We have revised the Discussion to clarify that marital status was significant in bivariate analysis but was not retained as an independent predictor in the adjusted multivariable model.

---

## [Editor Report · Decision Letter 2]

10 Mar 2026

Prevalence and risk factors of disordered eating risk among Yemeni medical students: a national multicenter cross-sectional study

PONE-D-25-61542R2

Dear Dr. sharyan,

We’re pleased to inform you that your manuscript has been judged scientifically suitable for publication and will be formally accepted for publication once it meets all outstanding technical requirements.

Kind regards,

Mohammed Zawiah

Academic Editor

PLOS One
---

## [Editor Report · Acceptance letter]

PONE-D-25-61542R2

PLOS One

Dear Dr. sharyan,

I'm pleased to inform you that your manuscript has been deemed suitable for publication in PLOS One. Congratulations! Your manuscript is now being handed over to our production team.

Kind regards,

on behalf of

Dr. Mohammed Zawiah

Academic Editor

PLOS One